# Design-Bench: Benchmarks for Data-Driven Offline Model-Based Optimization

**Brandon Trabucco, Xinyang Geng, Aviral Kumar, Sergey Levine**
Electrical Engineering and Computer Sciences, University of California Berkeley.

## Abstract

Black-box model-based optimization (MBO) problems, where the goal is to find a design input that maximizes an unknown objective function, are ubiquitous in a wide range of domains, such as the design of proteins, DNA sequences, aircraft, and robots. Solving model-based optimization problems typically requires actively querying the unknown objective function on design proposals, which means physically building the candidate molecule, aircraft, or robot, testing it, and storing the result. This process can be expensive and time consuming, and one might instead prefer to optimize for the best design using only the data one already has. This setting—called offline MBO—poses substantial and different algorithmic challenges than more commonly studied online techniques. A number of recent works have demonstrated success with offline MBO for high-dimensional optimization problems using high-capacity deep neural networks. However, the lack of standardized benchmarks in this emerging field is making progress difficult to track. To address this, we present Design-Bench, a benchmark for offline MBO with a unified evaluation protocol and reference implementations of recent methods. Our benchmark includes a suite of diverse and realistic tasks derived from real-world optimization problems in biology, materials science, and robotics that present distinct challenges for offline MBO. Our benchmark and reference implementations are publicly available at: [github.com/brandontrabucco/design-bench](github.com/brandontrabucco/design-bench)

## 1 Introduction

Automatically synthesizing designs that maximize a desired objective function is one of the most important challenges in scientific and engineering disciplines. From protein design in molecular biology [33] to superconducting material discovery in physics [16], researchers have made significant progress in applying machine learning to optimization problems over structured design spaces. Commonly, the exact form of the objective function is unknown, and the objective value for a novel design can only be found by either running computer simulations or real world experiments. This process of optimizing an unknown function by only observing samples from this function is known as black-box optimization, and is typically solved in an **online** iterative manner, where in each iteration the solver proposes new designs and queries the objective function for feedback in order to inform better design proposals at the next iteration [42]. In many domains however, the objective function is prohibitively expensive to evaluate because it requires manually conducting experiments in the real world. In this setting, one cannot query the true objective function, and cannot receive feedback on design proposals. Instead, a collection of past records of designs and corresponding objective values might be available, and the optimization method must instead leverage existing data to synthesize the most optimal designs. This is the setting of **offline model-based optimization** (offline MBO).

Although online black-box optimization has been studied extensively, offline MBO has received comparatively less attention, and only a small number of recent works study offline MBO in the setting with high-dimensional design spaces [7, 22, 10, 11, 38]. This is partly because online techniques

Submitted to the 35th Conference on Neural Information Processing Systems (NeurIPS 2021) Track on Datasets and Benchmarks. Do not distribute.

cannot be directly applied in settings where offline MBO is used, especially in high-dimensional settings. Online techniques, such as Bayesian optimization [35], often require iterative feedback via queries to the objective function. Such online optimizers exhibit optimistic behavior: they rely on active queries at completely unseen designs irrespective of whether such a design is good or not. When access to these queries is removed, certain considerations change: optimism is no longer desirable and distribution shift becomes a major challenge [22].

Even with only a few existing offline MBO methods, it is hard to compare and track progress, as methods are typically proposed and evaluated on different tasks with distinct evaluation protocols. To the best of our knowledge, there is no commonly adopted benchmark for offline MBO. To address, we introduce a suite of tasks for offline MBO with a standardized evaluation protocol. We include a diverse set of tasks that span a wide range of application domains—from synthetic biology to robotics–that aims at representing the core challenges in real-world offline MBO. While the tasks are not intended to directly enable solving the corresponding real-world problems, which would require a lot of machinery in real hardware setup (e.g., a real robot or access to a wetlab for molecule design), they are intended to provide algorithm designers with a representative sampling of challenges that reflect the difficulties with real-world MBO. That is to say, the tasks are not intended to be *real*, but are intended to be *realistically challenging*. Further, the diversity of the tasks measures how they generalize across multiple domains and verifies they are not specialized to a single task. Our benchmark incorporates a variety of challenging factors, such as high dimensionality and sensitive discontinuous objective functions, which help identify the strengths and weaknesses of MBO methods. Along with this benchmark suite, we present reference implementations of a number of existing offline MBO and baseline optimization methods. We systematically evaluate them on all of the proposed benchmark tasks and report results. We hope that our work can provide insight into the progress of offline MBO methods and serve as a meaningful metric to galvanize research in this area.

## 2 Offline Model-Based Optimization Problem Statement

In online model-based optimization, the goal is to optimize a (possibly stochastic) black-box objective function $f(\mathbf{x})$ with respect to its input. The objective can be written as $\arg\max_{\mathbf{x}} f(\mathbf{x})$. Methods for online MBO typically optimize the objective iteratively, proposing design $\mathbf{x}_k$ at the $k$th iteration and query the objective function to obtain $f(\mathbf{x}_k)$. Unlike its online counterpart, access to the true objective $f$ is not available in offline MBO. Instead, the algorithm $\mathfrak{A}$ is provided access to a static dataset $\mathcal{D} = \{(\mathbf{x}_i, y_i)\}$ of designs $\mathbf{x}_i$ and a corresponding measurement of the objective value $y_i$. The algorithm consumes this dataset and produces an optimized candidate design $\mathbf{x}^*$ which is evaluated against the true objective function. Abstractly, the objective for offline MBO is:

$$\arg\max_{\mathfrak{A}} f(\mathbf{x}^*) \text{ where } \mathbf{x}^* = \mathfrak{A}(\mathcal{D}). \tag{1}$$

In practice, producing a single optimal design entirely from offline data is very difficult, so offline MBO methods are more commonly evaluated [22] in terms of "$P$ percentile of top $K$" performance, where the algorithm produces $K$ candidates and the $P$ percentile objective value determines final performance. Next we discuss two important aspects pertaining to offline MBO, namely, why offline MBO algorithms can improve beyond the best design observed in the offline dataset despite no active queries, and the associated challenges with devising offline MBO algorithms.

**Would offline MBO even produce designs better than the best observed design in the dataset?** A natural question to ask is whether it is even reasonable to expect offline MBO algorithms to improve over the performance of the best design seen in the dataset. As we will show in our benchmark results, many of the tasks that we propose do already admit solutions from existing algorithms that exceed the performance of the best sample in the dataset. To provide some intuition for how this can be possible, consider a simple example of offline MBO problems, where the objective function $f(\mathbf{x})$ can be represented as a sum of functions of independent partitions of the design variables,

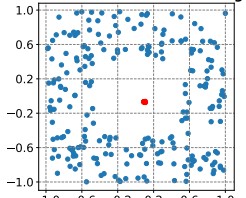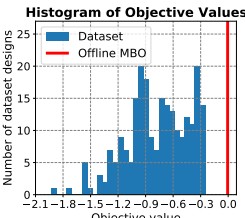

Figure 1: Offline MBO finds designs better than the best in the observed dataset by exploiting compositional structure of the objective function. **Left:** datapoints in a toy quadratic function MBO task over 2D space with optimum at $(0.0, 0.0)$ in blue, MBO found design in red. **Right:** Objective value for optimal design is much higher than that observed in the dataset.

| Dataset Name | ‖ | Size | Cardinality | Categories | Type | Oracle |
|---|---|---|---|---|---|---|
| **TF Bind 8** | | 65792 | 8 | 4 | Discrete | Exact |
| **GFP** | | 56086 | 237 | 20 | Discrete | Transformer |
| **UTR** | | 280000 | 50 | 4 | Discrete | Transformer |
| **ChEMBL** | | 40516 | 425 | 591 | Discrete | CNN |
| **Superconductor** | | 21263 | 86 | N/A | Continuous | Random Forest |
| **Hopper Controller** | | 3200 | 5126 | N/A | Continuous | Exact |
| **Ant Morphology** | | 25009 | 60 | N/A | Continuous | Exact |
| **D'Kitty Morphology** | | 25009 | 56 | N/A | Continuous | Exact |

Table 1: **Overview of the tasks in our benchmark suite.** Design-Bench includes a variety of tasks from different domains with both discrete and continuous design spaces and 3 high-dimensional tasks with $> 200$ design dimensions, making it suitable for benchmarking offline MBO methods.

i.e., $f(\mathbf{x}) = f_1(\mathbf{x}[1]) + f_2(\mathbf{x}[2]) + \cdots + f_N(\mathbf{x}[N]))$, where $\mathbf{x}[1], \cdots, \mathbf{x}[N]$ denotes disjoint subsets of design variables $\mathbf{x}$. The dataset of the offline MBO problem contains optimal design variable for each partition, but not the combination. If an offline MBO algorithm can identify the compositional structure of independent partitions, it would be able to combine the optimal design variable for each partition together to form the overall optimal design and therefore improving the performance over the best design in the dataset. To better demonstrate this idea, we created a toy problem in two dimensions, where the objective function is simply $f(x, y) = -x^2 - y^2$. We collect a dataset of uniformly sampled $x$ and $y$ from $-1$ to 1, but discard the samples that have the combination of best $x$ and $y$. We then run a naïve gradient ascent algorithm, as we will describe later in this paper. In Figure 1, we can clearly see that our offline MBO algorithm is able to learn to combine the best $x$ and $y$ and produce designs significantly better than the best sample in the dataset. Such a condition appears in a number of scenarios in practice e.g., in reinforcement learning (RL), where the Markov structure provides a natural decomposition satisfying this composition criterion [12] and effective offline RL algorithms are known to exploit this structure [12] or in protein design, where objective such as fluorescence naturally decompose into functions of neighboring Amino acids [7].

**What makes offline MBO especially challenging?** The offline nature of the problem prevents the algorithm $\mathfrak{A}$ from querying the ground truth objective $f$ with its proposed design candidates, and this makes the offline MBO problem much more difficult than the online design optimization problem. One naïve approach to tackle this problem is to learn a model of the objective function using the dataset, which we can denote $\hat{f}(\mathbf{x})$, and then convert this offline MBO problem into a regular online MBO problem by treating the learned objective model as the true objective. However, this generally does not work: optimizing the design $\mathbf{x}$ with respect to a learned proxy $\hat{f}(\mathbf{x})$ will produce *out-of-distribution* designs that "fool" $\hat{f}(\mathbf{x})$ into outputting a high value, analogously to adversarial examples. Indeed, it is well known that optimizing naïvely with respect to model inputs to obtain a desired output will usually simply "fool" the model [22]. A naïve strategy to address this out-of-distribution issue is to constrain the design to stay close to the data, but this is also problematic, since in order to produce a design that is better than the best training point, it is usually necessary to deviate from the training data at least somewhat. In almost all practical MBO problems, such as optimization over proteins or robot morphologies as we discuss in section 5, designs with the highest objective values typically lie on the tail of the dataset distribution and we may not find them by staying extremely close to the data distribution. This conflict between the need to remain close to the data to avoid out-of-distribution inputs and the need to deviate from the data to produce better designs is one of the core challenges of offline MBO. This challenge is often exacerbated in real-world settings by the high dimensionality of the design space and the sparsity of the available data, as we will show in our benchmark. A good offline MBO method needs to carefully balance these two sides, producing optimized designs that are good, but not too far from the data distribution.

## 3 Related Work

Prior work has extensively focused on online or active MBO which requries active querying on the ground truth function, including algorithms using Bayesian optimization and their scalable variants [23, 35, 36, 32, 26], direct search [21], genetic or evolutionary algorithms [41, 25, 45], cross-entropy method [28], simulated annealing [39], etc. While efficient in solving the optimization problem if the ground truth function can be easily evaluated, these methods are not well suited for real-world problems where the ground truth function is expensive to evaluate and therefore prohibitive for active querying. On the other hand, offline MBO that only utilizes an already existing database of designs and objective values, for example, those obtained via previous experiments, presents an

attractive algorithmic paradigm towards approaching such scenarios. Since offline MBO prohibits any ability to query the groundtruth objective actively, offline MBO presents different challenges from the typically studied online MBO problem as we discuss in Section 5. We believe that these challenges push the need for a new set of benchmarks to properly evaluate offline MBO methods.

The most important components for a good offline MBO benchmark are datasets that capture the challenges of real-world problems. Fortunately, researchers working on a wide variety of scientific fields have already collected many datasets of designs which we can use for training offline MBO algorithms. Sarkisyan et al. [30] analyze the fluorescence of GFP proteins under blue and ultraviolet light, and Brookes et al. [7] use this dataset for optimization to find the protein with the highest fluorescence value. ChEMBL [13] provides a dataset for drug discovery, where molecule activities are measured against a target assay. Hamidieh [16] analyze the critical temperatures for superconductors and provide a dataset to search for room-temperature superconductors with potential in the construction of quantum computers. Some of these datasets have already been employed in the study of offline MBO methods [22, 7, 10]. However, these studies all use different sets of tasks and their evaluation protocols are highly domain-specific, making it difficult to compare across methods. In our benchmark, we incorporate modified variants of some of these datasets along with our own tasks and provide a standardized evaluation protocol. We hope that these tasks can represent realistic MBO problems across a wide range of domains and that the standardized evaluation protocol can facilitate development of new and more powerful offline MBO algorithms.

Recently, several methods have been proposed for specifically addressing the offline MBO problem. These methods [22, 7, 10] typically learn models of the objective function and optionally, a generative model [20, 14, 24] of the design manifold and use them for optimization. We discuss these methods in detail in Section 6 and benchmark their performance in Section 7.

# 4 Design-Bench Benchmark Tasks

In this section, we describe the set of tasks included in our benchmark. An overview of the tasks is provided in Table 1. Each task in our benchmark suite comes with a dataset $\mathcal{D} = \{(\mathbf{x}_i, y_i)\}$, along with a ground-truth oracle objective function $f$ that can be used for evaluation. An offline MBO algorithm should not query the ground-truth oracle function during training, even for hyperparameter tuning. We first discuss the nature of oracles used in Design-Bench.

**Expert model as oracle function.** While in some of the tasks in our benchmark, such as tasks pertaining to robotics (Hopper Controller, D'Kitty Morphology, and Ant Morphology), the oracle functions are evaluated by running computer simulations to obtain the true objective values, in the other tasks, the true objective values can only be obtained by conducting expensive physical experiments. While the eventual aim of offline MBO is to make it possible to optimize designs in precisely such settings, requiring real physical experiments for evaluation makes the design and benchmarking of new algorithms difficult and time consuming. Therefore, to facilitate benchmarking, we follow the evaluation methodology in prior work [7, 10] and use models built by domain experts as our ground-truth oracle functions. Note, however, that the training data provided for offline MBO is still real data – the domain expert model is used *only* to evaluate the result for benchmarking purposes. In many cases, these expert models are *also* learned, but with representations that are hand-designed, with built-in domain-specific inductive biases. The ground-truth oracle models are also trained on much more data than is made available for solving the offline MBO problem, which increases the likelihood that this expert model can provide an accurate evaluation of solutions found by offline MBO, even if they lie outside the training distribution. While this approach to evaluation diminishes the realism of our benchmark since these proxy "true functions" may not always be accurate, we believe that this trade off is worthwhile to make benchmarking practical. The main purpose of our benchmark is to facilitate the evaluation and development of offline MBO algorithms, and we believe that it is important to include tasks in domains where the true objective values can only be obtained via physical experiments, which make up a large portion of the real-world MBO problems. We provide further analysis of the fidelity of our expert model oracles in Appendix F.

We now provide a detailed description of the tasks in our benchmark. A description of the data collection strategy and the data pre-processing strategy can be found in Appendix A.

**GFP: protein fluorescence maximization.** The goal of this task is to design a derivative protein from the *Aequorea victoria* green fluorescent protein (GFP) that has maximum fluorescence, using a real-world dataset mapping proteins to fluorescence collected by Sarkisyan et al. [30]. While we

cannot precisely evaluate any novel protein, we employ an expert Transformer regression model [27] as the oracle function, following the convention in prior work [7, 10]. Our Transformer is trained on the complete GFP dataset containing 56,086 proteins and corresponding fluorescence values. The model achieves a final Spearman's rank-correlation coefficient with a held-out validation set of 0.8497. The design space is discrete, consisting of sequences of 237 categorical variables that take one of 20 options, which corresponds to a sequence of amino acids.

**TF Bind 8 and UTR: DNA sequence optimization.** The goal of TF Bind 8 is to find the length-8 DNA sequence with maximum binding affinity with a particular transcription factor (`SIX6_REF_R1` by default). The ground truth binding affinities for all 65,792 designs are available [5]. The goal of the UTR task is to find a human length-50 5'UTR DNA sequence that maximizes the expression level of its corresponding gene. Following Sample et al. [29], we train a Transformer oracle to predict ribosome load from length-50 DNA sequences. The Transformer is trained on the entire UTR dataset used in Sample et al. [29], consisting of 280,000 DNA sequences and measured ribosome loads. The oracle achieves a final Spearman's rank-correlation with a held-out validation set of 0.8617. The design space consists of sequences of one of four categorical variables, one for each nucleotide.

**ChEMBL: molecule design via SMILES [40] strings.** This task is taken from the domain of drug discovery with the goal to design the SMILES [40] string of a molecule that exhibits high activity with a target assay. We adapt the ChEMBL [13] dataset and choose the standard type `GI50` and ASSAY_ChEMBL_ID `CHEMBL1964047`, resulting in a dataset of 40,516 pairs of SMILES strings and `GI50` values. The true `GI50` value can only be determined by physical experiments, so we train a convnet oracle to predict `GI50` values from SMILES on all 40,516 examples, which achieves a final Spearman's rank-correlation on a held-out validation set of 0.3208. The design space is a sequence of 425 categorical variables that take any of 591 options, representing tokenized SMILES strings.

**Superconductor: critical temperature maximization.** The Superconductor task is taken from the domain of materials science, where the goal is to design the chemical formula for a superconducting material that has a high critical temperature. We adapt a real-world dataset proposed by Hamidieh [16]. The dataset contains 21263 superconductors annotated with critical temperatures. Prior work has employed this dataset for the study of offline MBO methods [10], and we follow their convention using a random forest regression model, detailed in [16], for our oracle. The model achieves a final Spearman's rank-correlation coefficient with a held-out validation set of 0.9210. The design space for Superconductor is a vector with 86 real-valued components representing the mixture of elements by number of atoms in the chemical formula of each superconductor.

**Ant and D'Kitty Morphology: robot morphology optimization.** The goal is to optimize the morphological structure of two simulated robots: Ant from OpenAI Gym [6] and D'Kitty from ROBEL [1]. For Ant Morphology, the we need to optimize the morphology of a quadruped robot, to run as fast as possible, with a pre-trained neural network controller. For D'Kitty Morphology, the goal is to optimize the morphology of D'Kitty robot (shown on the right), such that a pre-trained neural network controller can navigate the robot to a 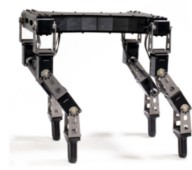

fixed location. Thus the goal is to find morphologies optimal for the pre-trained controller. The pre-trained neural network controller is a morphology conditioned action predictor trained to work well on a large rage of morphologies. The morphology parameters of both robots include size, orientation, and location of the limbs, giving us 60 continuous values in total for Ant and 56 for D'Kitty. To evaluate a given design, we run robotic simulation in MuJoCo [37] for 100 time steps, averaging 16 independent trials giving us reliable but cheap to compute estimates.

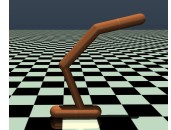 **Hopper Controller: robot neural network controller optimization.** The goal in this task is to optimize the weights of a neural network policy so as to maximize the expected discounted return on the Hopper-v2 locomotion task in OpenAI Gym [6]. While this might appear similar to reinforcement learning (RL), our formulation is distinct: unlike RL, we don't have access to any form of trajectory data in the dataset. Instead, our dataset only comprises of neural network controller weights and the corresponding return values, which invalidates the applicability of conventional RL methods. We evaluate the true objective value of any design by running 1000 steps of simulation in the MuJoCo simulator conventionally ussed with this environment. The design space of this task is high-dimensional with 5126 continuous variables corresponding to the flattened weights of a neural network controller. The dataset is collected by training a PPO [31] and recording the agent's weights every 10,000 samples.

## 5  Task Properties, Challenges, and Considerations

The primary goal of our proposed benchmark is to provide a general test bench for developing, evaluating, and comparing algorithms for offline MBO. While in principle any online active black-box optimization problem can be formulated into an offline MBO problem by collecting a dataset of designs and corresponding objective measurements, it is important to pick a subset of tasks that represent the challenges of real-world problems in order to convincingly evaluate algorithms and obtain insights about algorithm behavior. Therefore, several factors must be considered when choosing the tasks, which we discuss next.

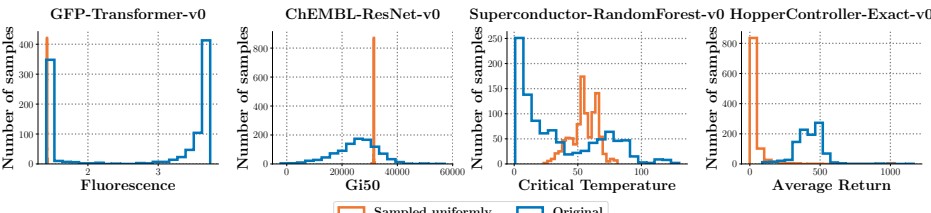

Figure 2: **Histogram (frequency distribution) of objective values in the dataset compared to a uniform re-sampling of the dataset** from the design space. In every case, re-sampling skews the distribution of values to the left, suggesting that there exists a thin manifold of valid designs in the high-dimensional design space, and most of the volume in this space is occupied by low-scoring designs. The distribution of objective values in the dataset are often heavy-tailed, for instance, in the case of ChEMBL and Superconductor.

**Diversity and realistically challenging.** First of all, the tasks need to be diverse and realistically challenging in order to prevent offline MBO algorithms from overfitting to a particular problem domain and to expect that methods performing well on this benchmark suite would also perform well on real-world offline MBO problems. Design-Bench consists of tasks that are diverse in many respects. It includes both tasks with *discrete* and with *continuous* design spaces. Continuous design spaces, equipped with metric space and ordering structures, could make the problem easier to solve than discrete design spaces. However, discrete design spaces are finite and therefore might enjoy better dataset coverage than some continuous tasks. A strong offline MBO algorithm needs to be able to handle both cases. Further, our tasks have varying dimensionality, ranging from 56 to 5126 dimensions. While our tasks are not intended to directly solve real-world problems (e.g., we don't actually expect the best robot morphology in our benchmark to actually correspond to the best possible real robot morphology), they are intended to provide method designers with a representative sampling of challenges that reflect the kinds of difficulties they would face with real-world datasets, making them realistically challenging.

**High-dimensional design spaces.** In many real-world offline MBO problems, such as drug discovery [13], the design space is *high-dimensional* and the valid designs sprasely lie on a *thin manifold* in this high-dimensional space. This property poses a unique challenge for many MBO methods: to be effective on such problem domains, MBO methods need to capture the thin manifold of the design space to be able to produce valid designs. Prior work [22] has noted that this can be very hard in practice. In our benchmark, we include GFP, ChEMBL and HopperController tasks with up to 5000 dimensional design spaces to capture this challenge. To intuitively understand this challenge, we performed a study on some tasks in Figure 2, where we sampled 3200 designs uniformly at random from the design space and plotted a histogram of the objective values against those in the dataset we provide, which only consists of valid designs. Observe the discrepancy in objective values, where randomly sampled designs generally attain objective values much lower than the dataset average. This indicates that valid designs only lie on a thin manifold in the design space and therefore we are very unlikely to hit a valid design by random sampling.

**Highly sensitive objective function.** Another important challenge that should be taken into consideration is the *high sensitivity* of objective functions, where closeness of two designs in design space need not correspond to closeness in their objective values, which may differ drastically. This challenge is naturally present in practical problems like protein synthesis [33], where the change of a single amino acid could significantly alter the property of the protein. The DKittyMorphology and AntMorphology tasks in our benchmark suite are also particularly challenging in this direction. To visualize the high sensitivity of the objective function, we plot a one dimensional slice of the objective function around a single sample in our dataset in Figure 3. Observe that with other variables kept the same, slightly altering one variable can significantly reduce the objective value, making it hard for offline MBO methods to produce the optimal design.

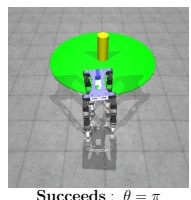 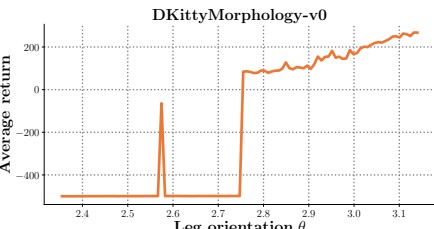 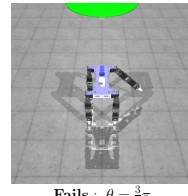

Figure 3: **Highly sensitive landscape of the ground truth objective function in DKittyMorphology.** A small change in a single dimension of the design space, for instance changing the orientation $\theta$ (x-axis) of the base of the robot's front right leg, critically impacts the performance value (y-axis). The robot's design is the original D'Kitty design and is held constant while varying $\theta$ uniformly from $\frac{3}{4}\pi$ to $\pi$.

**Heavy-tailed data distributions.** Finally, another challenging property for offline MBO methods is the shape of the data distribution. Learning algorithms are likely to exhibit poor learning behavior when the distribution of objective values in the dataset is heavy-tailed. This challenge is often present in black-box optimization [8] and can hurt the performance of MBO algorithms that use a generative model as well as those that use a learned model of the objective function. As shown in Figure 2 tasks in our benchmark exhibit this heavy-tailed structure.

## 6   Algorithm Implementations In Design-Bench

To provide a baseline for comparisons in future work, we benchmark a number of recently proposed offline MBO algorithms on each of our tasks. Since the dimensionality of our tasks ranges from 56 to 5126, we chose prior methods that can handle *both* the case of offline training data (i.e., no active interaction) and high-dimensional inputs. Thus, we include MINs [22], CbAS [7], autofocusing CbAS [10] and REINFORCE/CMA-ES [43] in our comparisons, along with a baseline naïve "gradient ascent" method that approximates the true function $f(\mathbf{x})$ with a deep neural network and then performs gradient ascent on the output of this model. In this section, we briefly discuss these algorithms, before performing a comparative evaluation in the next section. Our implementation of these algorithms are open sourced and can be found at github.com/brandontrabucco/design-baselines.

**Gradient ascent (Grad).** This is a simple baseline that learns a model of the objective function, $\hat{f}(\mathbf{x})$, and optimizes $\mathbf{x}$ against this learned model via gradient ascent. Formally, the optimal solution $\mathbf{x}^*$ generated by this method can be computed as a fixed point of the following update: $\mathbf{x}_{t+1} \leftarrow \mathbf{x}_t + \alpha \nabla_{\mathbf{x}} \hat{f}(\mathbf{x})|_{\mathbf{x}=\mathbf{x}_t}$. In practice we perform $T = 200$ gradient steps, and report $\mathbf{x}_T$ as the final solution. Such methods are susceptible to producing invalid solutions, since the learned model does not capture the manifold of valid-designs and hence cannot constrain the resulting $\mathbf{x}_T$ to be on the manifold. We additionally evaluate a variant (**Grad. Min**) optimizing over the minimum prediction of $N = 5$ learned objective functions in an ensemble of learned objective functions and (**Grad. Mean**) that optimizes the mean ensemble prediction. We discuss additional tricks (e.g., normalization of inputs and outputs) that we found beneficial with this baseline in Appendix D.

**Covariance matrix adaptation (CMA-ES).** CMA-ES Hansen [17] is a simple optimization algorithm that maintains a belief distribution over the optimal design, and gradually refines this distribution by adapting the covariance matrix using feedback from a (learned) objective function, $\hat{f}(\mathbf{x})$. Formally, let $\mathbf{x}_t \sim \mathcal{N}(\mu_t, \Sigma_t)$ be the samples obtained from the distribution at an iteration $t$, then CMA-ES computes the value of learned $\hat{f}(\mathbf{x}_t)$ on samples $\mathbf{x}_t$, and fits $\Sigma_{t+1}$ to the highest scoring fraction of these samples and repeats this multiple times. The learned $\hat{f}(\mathbf{x})$ is trained via supervised regression.

**REINFORCE [43].** We also evaluated a method that optimizes a learned objective function, $\hat{f}(\mathbf{x})$, using the REINFORCE-style policy-gradient estimator. REINFORCE is capable of handling non-smooth and highly stochastic objectives, making it an effective choice. This method parameterizes a distribution $\pi_\theta(\mathbf{x})$ over the design space and then updates the parameters $\theta$ of this distribution towards the design that maximizes $\hat{f}(\mathbf{x})$, using the gradient, $\mathbb{E}_{\mathbf{x} \sim \pi_\theta(\mathbf{x})}[\nabla_\theta \log \pi_\theta(\mathbf{x}) \cdot \hat{f}(\mathbf{x})]$. We train an ensemble of $\hat{f}(\mathbf{x})$ models and pick the subset of models that satisfy a validation loss threshold $\tau$. This threshold is task-specific; for example, $\tau \le 0.25$ is sufficient for Superconductor-v0.

**Conditioning by adaptive sampling (CbAS) [7].** CbAS learns a density model in the space of design inputs, $p_0(\mathbf{x})$ that approximates the data distribution and gradually adapts it towards the optimized solution $\mathbf{x}^*$. In a particular iteration $t$, CbAS alternates between **(1)** training a variational auto-encoder (VAE) [20] on a set of samples generated from the previous model $\mathcal{D}_t = \{\mathbf{x}_i\}_{i=1}^m; \mathbf{x}_i \sim$

$p_{t-1}(\cdot)$ using a weighted version of the standard ELBO objective biased towards *estimated* better designs and **(2)** generating new design samples from the autoencoder to serve as $\mathcal{D}_{t+1} = \{\mathbf{x}_i | \mathbf{x}_i \sim p_t(\cdot)\}$. In order to estimate the objective values for designs sampled from the learned density model $p_t(\mathbf{x})$, CbAS utilizes separately trained models of the objective function, $\hat{f}(\mathbf{x})$ trained via supervised regression. This training process, at a given iteration $t$, is:

$$p_{t+1}(\mathbf{x}) := \arg\min_p \frac{1}{m} \sum_{i=1}^m \frac{p_0(\mathbf{x}_i)}{p_t(\mathbf{x}_i)} P(\hat{f}(\mathbf{x}_i) \geq \tau) \log p_t(\mathbf{x}_i)$$

$$\text{where } \{\mathbf{x}_i\}_{i=1}^m \sim p_t(\cdot). \tag{2}$$

**Autofocused CbAS (Auto. CbAS) [10].** Since CbAS uses a learned model of the objective function $\hat{f}(\mathbf{x})$ to iteratively adapt the generative model $p(\mathbf{x})$ towards the optimized design, the function $\hat{f}(\mathbf{x})$ will inevitably be required to make predictions on shifting design distributions $p_t(\mathbf{x})$. Hence, any inaccuracy in these values can adversely affect the optimization procedure. Autofocused CbAS aims to correct for this shift by re-training $\hat{f}(\mathbf{x})$ (now denoted $\hat{f}_t(\mathbf{x})$) under the design distribution given by the current model, $p_t(\mathbf{x})$ via importance sampling, which is then fed into CbAS.

$$\hat{f}_{t+1} := \arg\min_{\hat{f}} \frac{1}{|\mathcal{D}|} \sum_{i=1}^{|\mathcal{D}|} \frac{p_t(\mathbf{x}_i)}{p_0(\mathbf{x}_i)} \cdot \left(\hat{f}(\mathbf{x}_i) - y_i\right)^2,$$

**Model inversion networks (MINs) [22].** MINs learn an inverse map from the objective value to a design, $\hat{f}^{-1} : \mathcal{Y} \to \mathcal{X}$ by using objective-conditioned inverse maps, search for optimal $y$ values during optimization and finally query the learned inverse map to produce the corresponding optimal design. MIN minimizes a divergence measure $\mathcal{L}_p(\mathcal{D}) := \mathbb{E}_{y \sim p_\mathcal{D}(y)} \left[ D(p_\mathcal{D}(\mathbf{x}|y), \hat{f}^{-1}(\mathbf{x}|y)) \right]$ to train such an inverse map. During optimization, MINs obtain the optimal $y$-value that is used to query the inverse map, and obtains the optimized design by sampling form the inverse map.

**Bayesian optimization (BO-qEI).** We perform offline Bayesian optimization to maximize the value of a learned objective function, $\hat{f}(\mathbf{x})$, by fitting a Gaussian Process, proposing candidate solutions, and labeling these candidates using $\hat{f}(\mathbf{x})$. To improve efficiency, we choose the quasi-Expected-Improvement acquisition function [44], and the implementation from the BoTorch framework [4].

## 7 Benchmarking Prior Methods

In this section, we provide a comparison of prior algorithms discussed in Section 6 on our proposed tasks. For purposes of standardization, easy benchmarking, and future algorithm development, we present results for all Design-Bench tasks in Table 2. As discussed in Section 2, we provide each method with a dataset, and allow it to produce $K = 128$ optimized design candidates. These $K = 128$ candidates are then evaluated with the oracle function, and we report the $100^{\text{th}}$ percentile performance among them averaged over 8 independent runs, following convention in prior offline MBO work [10, 7, 22]. We also provide unofmralized and $50^{\text{th}}$%ile results in Appendices C.3, C.2.

**Algorithm setup and hyperparameter tuning.** Since our goal is to generate high-performing solutions without *any* knowledge of the ground truth function, any form of hyperparameter tuning on the parameters of the learned model should crucially respect this evaluation boundary and tuning must be performed completely offline, agnostic of the objective function. We provide a recommended method for tuning each algorithm described in Section 6 in Appendix E, which also serves as a set of guidelines for tuning future methods with similar components.

To briefly summarize, **for CbAS**, hyperparameter tuning amounts to finding a stable configuration for a VAE, such that samples from the prior distribution map to on-manifold designs after reconstruction. We empirically found that a $\beta$-VAE was essential for stability of CbAS—and high values of $\beta > 1$ are especially important for modelling high-dimensional spaces like that of HopperController. As a general task-agnostic principle for selecting $\beta$, we choose the smallest $\beta$ such that the VAE's latent space does not collapse during importance sampling. Collapsing latent-spaces seem to coincide with diverging importance sampling, and the VAE's reconstructions collapsing to a single mode. **For MINs**, hyperparameter tuning amounts to fitting a good generative model. We observe that MINs is particularly sensitive to the scale of $y_i$ when conditioning, which we resolve by normalizing the objective values. We implement MINs using WGAN-GP, and find that similar hyperparameters work well-across domains. **For Gradient Ascent**, while prior work has generally obtained extremely poor performance for naïve gradient ascent based optimization procedures on top of learned models of

the objective function, we find that by normalizing the designs $\mathbf{x}$ and objective values $y$ to have unit Gaussian statistics, and by multiplying the learning rate $\alpha \leftarrow \alpha\sqrt{d}$ where $d$ is the dimension of the design space (discussed in Appendix D), a naïve gradient ascent based procedure performs reasonably well on most tasks without task-specific tuning. For discrete tasks, only the objective values are normalized, and optimization is performed over log-probabilities of designs. We then uniformly evaluate samples obtained by running 200 steps of gradient ascent starting from the top scoring 128 samples in each dataset. Tuning instructions for each baseline are available in Appendix E.

**Results.** The results for all tasks are provided in Table 2. There are several takeaways from these results. First, these results indicate that there is no clear winner between the three prior offline MBO methods (MINs, CbAS, and Autofocused CbAS), provided they are all trained offline with no access to ground truth evaluation for any form of hyperparameter tuning. Furthermore, perhaps somewhat surprisingly, a naïve gradient ascent baseline is competitive

| | | GFP | TF Bind 8 | UTR | ChEMBL |
|---|---|---|---|---|---|
| Auto. CbAS | | 0.865 ± 0.000 | 0.910 ± 0.044 | 0.650 ± 0.006 | 0.470 ± 0.000 |
| CbAS | | 0.865 ± 0.000 | 0.927 ± 0.051 | 0.650 ± 0.002 | 0.517 ± 0.055 |
| BO-qEI | | 0.254 ± 0.352 | 0.798 ± 0.083 | 0.659 ± 0.000 | 0.333 ± 0.035 |
| CMA-ES | | 0.054 ± 0.002 | 0.953 ± 0.022 | 0.666 ± 0.004 | 0.350 ± 0.017 |
| Grad. | | 0.864 ± 0.001 | 0.977 ± 0.025 | 0.639 ± 0.009 | 0.360 ± 0.029 |
| Grad. Min | | 0.864 ± 0.000 | 0.984 ± 0.012 | 0.647 ± 0.007 | 0.361 ± 0.004 |
| Grad. Mean | | 0.864 ± 0.000 | 0.986 ± 0.012 | 0.647 ± 0.005 | 0.373 ± 0.013 |
| MINs | | 0.865 ± 0.001 | 0.905 ± 0.052 | 0.649 ± 0.004 | 0.473 ± 0.057 |
| REINFORCE | | 0.865 ± 0.000 | 0.948 ± 0.028 | 0.646 ± 0.005 | 0.459 ± 0.036 |
| | | Superconductor | Ant Morphology | DKitty Morphology | Hopper Controller |
| Auto. CbAS | | 0.421 ± 0.045 | 0.884 ± 0.046 | 0.906 ± 0.006 | 0.137 ± 0.005 |
| CbAS | | 0.503 ± 0.069 | 0.879 ± 0.032 | 0.892 ± 0.008 | 0.141 ± 0.012 |
| BO-qEI | | 0.402 ± 0.034 | 0.820 ± 0.000 | 0.896 ± 0.000 | 0.550 ± 0.118 |
| CMA-ES | | 0.465 ± 0.024 | 1.219 ± 0.738 | 0.724 ± 0.001 | 0.604 ± 0.215 |
| Grad. | | 0.518 ± 0.024 | 0.291 ± 0.023 | 0.874 ± 0.022 | 1.035 ± 0.482 |
| Grad. Min | | 0.506 ± 0.009 | 0.478 ± 0.064 | 0.889 ± 0.011 | 1.391 ± 0.589 |
| Grad. Mean | | 0.499 ± 0.017 | 0.444 ± 0.081 | 0.892 ± 0.011 | 1.586 ± 0.454 |
| MINs | | 0.469 ± 0.023 | 0.916 ± 0.036 | 0.945 ± 0.012 | 0.424 ± 0.166 |
| REINFORCE | | 0.481 ± 0.013 | 0.263 ± 0.032 | 0.562 ± 0.196 | -0.020 ± 0.067 |

Table 2: **100th percentile** evaluations. Results are averaged over 8 trials, and ± indicates the standard deviation of the reported objective value. For a description of the objective normalization methodology, please refer to Appendix C.1. *The MINs result for ChEMBL is missing because the MINs architecture does not fit into our computational budget. We will update our GitHub when the result is ready.

with several highly sophisticated MBO methods in 4 out of 8 tasks (Table 2), especially on high-dimensional tasks (e.g., HopperController). This result suggests that it might be difficult for generative models to capture high-dimensional task distributions with enough precision to be used for optimization, and in a number of tasks, these components might be unnecessary. However, on the other hand, as described in Appendix D and E.4, this simple baseline is also sensitive to certain design choices such as input normalization schemes and the number of optimization steps $T$. Therefore, while not a full-fledged offline MBO method, we believe that gradient ascent has potential to form a fundamental building block for future offline MBO methods. Finally, we remark that the performance of methods in Table 2 differ from the those reported by prior works. This difference stems from the standardization procedure employed in dataset generation (which we discuss in Appendix A), and the use of task-agnostic, uniform hyperparameter tuning.

## 8 Discussion and Conclusion

Offline MBO carries the promise to convert existing databases of designs into powerful optimizers, without the need for expensive real-world experiments for actively querying the ground truth objective function. However, due to the lack of standardized benchmarks and evaluation protocols, it has been difficult to accurately track the progress of offline MBO methods. To address this problem, we introduce Design-Bench, a benchmark suite of offline MBO tasks that covers a wide variety of domains, and both continuous and discrete, low and high dimensional design spaces. We provide a comprehensive evaluation of existing methods under identical assumptions. The comparatively high efficacy of even simple baselines such as naïve gradient ascent suggests the need for careful tuning and standardization of methods in this area. An interesting avenue for future work in offline MBO is to devise methods that can be used to perform model-selection and hyperparameter selection. One approach to address this problem is to devise methods for offline evaluation of produced solutions, which is also an interesting topic for future work. We hope that our benchmark will be adopted as the standard metric in evaluating offline MBO algorithms and provides insight in future algorithm development. Since our benchmark aims to standardize the evaluation of offline MBO, we note that while it may have both positive (e.g., enhancing human life quality via automation) and negative (e.g., loss of jobs) impact on society, all these impacts are more broadly applicable to offline MBO algorithms in general and not specifically to this work.

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
