# OpenReview forum: "Design-Bench: Benchmarks for Data-Driven Offline Model-Based Optimization"
_NeurIPS.cc/2021/Track/Datasets_and_Benchmarks/Round1 — Submitted to NeurIPS 2021 Datasets and Benchmarks Track (Round 1)_

### Official Review · Reviewer_TorJ · 2021-07-01
**A good paper in need of structural revision**

**Rating:** 5
**Confidence:** 4

**Strengths:**

This paper has three main strengths; the paper is well-motivated, well-written, and very thorough.

The authors do an excellent job at describing the field and motivating the problem. They take great care to describe the many disparate and complex peculiarities of each task/dataset, oracle, and algorithm. (I do posit below that there is slightly too much detail on these specifics for the benchmark track. However, I'm confident that the authors can strike a balance for this venue.) I also enjoyed the space in the paper given to the considerations of what important characteristics are necessary for a benchmark in this field.

The authors are excellent communicators and the manuscript reads very well.

**Weaknesses:**

I see three main weaknesses in this paper:

Main weaknesses:

(WM1) The structure of the presentation of this benchmark is not yet sufficient for the NeurIPS dataset and benchmark track. See my comments in clarity for more on this. Essentially, I find the balance of motivation and benchmark methodology to be incorrect for a benchmark specific track. I'd like to see clarity and space given to the ways in which the evaluation is conducted. The ultimate goal of this benchmark is to (a) unify the community around the metrics and evaluation protocol presented, and (b) allow for further extension with new tasks/datasets/oracles and (most importantly) algorithms. I suggest the authors provide more details on the specifics of the evaluation methods and abstraction for the inclusion of a new task/dataset, oracle, and algorithm. Additionally, since a benchmark is only as good as its evaluation procedures, a subsection in the main paper which strongly advocates for your chosen procedures and what differences (in results) one might expect from different choices. For instance, if a different oracle is chosen, how much does that change the result?

(WM2) The only specific benefit detailed in this paper is the result regarding naive gradient ascent. Assuming that this is the type of result that the authors think validates the importance of this benchmark, the manuscript would seriously benefit from a more detailed discussion of it. How do the authors think this result will impact the community? What about other types of results that are derived from extensions of this benchmark?

(WM3) The ultimate reason why this benchmark is important is because it enables new offline MBO algorithms to be developed with broader applicability across the burgeoning field. This is necessary because MBO has so many applications in diverse domains that siloing of advancements can be deleterious to the whole. However, this paper, as written, does not convince me that this ultimate goal is appropriate given the highly-domain specific nature of each task. Essentially, why would the pursuit of a general purpose algorithm that works well on many offline MBO tasks be a better thing for the community than a more domain-specific solution which others adapt towards their own goal?



Minor weakness:

(Wm1) With n=8, it is standard practice to use a t-value instead of a std deviation in the tables.

**Additional Feedback:**

If the authors choose to update this manuscript for this track, I would encourage them to keep the overall structure of the paper, as it very much addresses many of the important aspects of a benchmark. My main suggestion for light revision is to rebalance the content to be less heavy on the domain-specific information and heavier on the specifics of the benchmark procedures.


The following are some small typos/grammatical errors in the manuscript. They didn't influence my opinion at all.

L97: improving -> improve

L106: objective -> objectives

L365: unofmralized -> unnormalized

L971: is -> are


**Clarity:**

The paper is well written. One main concept is not defined: 100 and 50th percentile performance. The meaning is not self-contained in the manuscript.

Looking at the structure of the manuscript, the majority is devoted to setting up the problem and motivation, and descriptions of the various tasks. The description of the evaluation methods (standardisation, evaluation protocol, etc) and results are relegated to either the appendix or (what feels like) a small and fast discussion at the end of the main paper in Section 7. Of course benchmark papers are very complex and require much more detail than could reasonably fit in the length of a NeurIPS submission. The authors made a reasonable choice to handle this inherent challenge.

I would have preferred more space in the main paper spent discussing the specific choices of the benchmarking procedure, namely the normalisation protocol (both score and gradient ascent), hyperparameter workflow, etc. As currently presented, the benchmark results do not feel self contained and many of the important considerations when evaluating a benchmark are relegated to the appendix or entirely missing. While always a challenge in motivating and distilling large and complex work, I think the reader would have benefitted from this description more than they would from the discussion of the tasks and design consideration motivations in Sections 4, 5, and 6.


**Correctness:**

Major point:

(CM1) Most evaluation decisions appear correct, though I believe that more discussion should be included in the main body of the paper to motivate some of the evaluation and design choices.

Minor points:

(Cm1)Two claims seemed to be in contradiction: the asterisk in the data tables corresponding to MINs and ChEMBL (indicating it was out of your computational budget) and the claim of Appendix C.4 are incongruous.

(Cm2) Figure 2, second figure appears to not "skew the distribution of values to the left" as described in the caption (though the main point about the thin manifold holds regardless).


**Documentation:**

Much of the documentation and pre-processing steps included in Appendix A are references to other papers. Without more details about where in the paper to find the specifics of the pre-processing, it will make reproducibility a challenge for a newcomer to a particular dataset. However, it appears this issue is handled in the github repo (design-bench/design-bench/datasets/) where the data, presumably, are in the proper format for reproducibility. The codebase in general is well-structured and sufficiently useable on its face.

If accepted, I would recommend referencing in the repo or separately in the appendix, a script which could fully reproduce each of the data tables (Table 2-4) and any other main results of the paper. This best practice generally will make reproducibility issues very minimal. Even though fully running this script from start to finish would be impractical, it is helpful for reproduction.


**Ethics:**

The authors have one sentence in the entire paper about ethics -- that this work (which aims to *standardise* offline MBO evaluation) has ethics which align with the ethics of the larger MBO field. I agree with this statement. It is apparent from the manuscript that the meta-goal of this work is to *accelerate* the adoption and impact of MBO writ large. So, this work would likely shorten the available space for consideration of ethical debates, concerns, or differing opinions about the general field.

A nice to have in this paper would be an additional section in the appendix which outlines the current state of ethical opinions about MBO to demonstrate the authors are engaged in this dialogue.


**Relation To Prior Work:**

Based on my reading of the paper, this work is substantially different from prior work, and all pertinent references are included therein. The work primarily synthesises and benchmarks other offline MBO (of which there are not many) and other important baselines for comparison. In this latter category, the authors chose sufficiently relevant algorithms from a much broader and diverse landscape.


**Summary And Contributions:**

The authors of Design-Bench propose a scheme to benchmark offline model-based optimisation techniques which do not have access to the true objective function and only have some data from which they can optimise. This field is not as studied as other settings where the objective is available, but is much harder and potentially more impactful to a broader audience. The offline-MBO community generally lacks standardised methods for comparing performance of algorithms across different domains. This work takes a first step towards proposing a scheme for doing this. The authors choose relevant datasets and tasks comparing current methods against standard oracles.

---

> ### Author Response · Authors · 2021-07-15
> **Response to Reviewer TorJ**
>
> First of all we want to thank the reviewer for the detailed comments and constructive suggestions. We address the reviewer’s concerns below.
>
> ____
>
> **WM1 & Clarity concerns.** We already include a number of these details in the paper and appendix. The **algorithm agnostic evaluation procedure** is described in Sections 2 and 7, where any generic algorithm is abstracted into a function that ingests the task dataset and produces K design candidates These candidates are evaluated using the oracle and the 100th percentile performance is reported as the final performance. The **algorithm specific hyperparameter tuning process** is elaborately described in Appendix E, and these procedures also serve as guidelines for practitioners tuning new algorithms that use similar components as the algorithms we benchmark. Regarding the **extension of new tasks and new algorithms**, we already have included documentation about the process in our GitHub repo [design-bench](https://github.com/brandontrabucco/design-bench) and [design-baselines](https://github.com/brandontrabucco/design-baselines). Regarding the **effect of different oracles**, we have included a study of the agreement between the oracles in Appendix F, which shows that different oracles tend to agree with each other about the performance of algorithms. We would be happy to add/clarify more details in the paper if the reviewer has more suggestions and we are more than happy to modify the organization to move some of these details to the main paper in the final if the reviewer has suggestions for a particular re-organization.
>
> ____
>
> **WM2.** Regarding the naive gradient ascent results, we believe that the effectiveness of this simple baseline demonstrates the importance of having a systematic benchmark. In creating this benchmark, we are aiming at developing a common suite of tasks and a systematic method for evaluating offline MBO algorithms, rather than promoting any specific algorithm. Therefore, gradient ascent is just one simple baseline in our benchmark and we are not proposing it as a new algorithm.
>
> ____
>
> **WM3.** We have seen in many areas of machine learning applications, directly using domain-agnostic algorithms tuned on benchmark tasks on real problems may not be optimal for the exact domain that a practitioner cares about. Similarly, offline MBO algorithms that are domain-agnostic may not out-of-the-box work well on all possible domains. However, we expect that similar to the benchmarks in supervised learning, design-bench will be used to discover high-level solution strategies and starting points for domain-specific solutions. Infact, we believe that the diversity of tasks in Design-bench, both in terms of the task-specifics (dimensionality, data points, etc) and the domains (physical sciences, materials, robotics) makes it a suitable candidate for testing the generality of offline MBO algorithms, which increase the chance that a successful algorithm on our benchmark could also work on another domain.
>
> ____
>
> **Ethics.** Like many other optimization fields, if offline MBO algorithms are applied to optimize mis-specified or unethical objectives, the end result might be harmful for society. For example, a company can use offline MBO to optimize people’s addiction to social media, which could induce harm to the users. Therefore, we strongly believe that when offline MBO algorithms are applied in a real-world problem, the potential ethical consequence must be carefully reviewed. However, we note that offline MBO could be extremely powerful in optimizing objectives relevant to advancement of society such as discovering new cures, designing better materials, improving power efficiency and addressing climate change.

---

### Official Review · Reviewer_nw61 · 2021-07-01
**An interesting benchmark, but some designs are questinable**

**Rating:** 5
**Confidence:** 4
**Correctness:** The design of the oracle function see…
**Clarity:** The paper is well-structured with cle…

**Strengths:**

1.	Black-box model-based optimization is an interesting and general problem. Offline black-box model-based optimization addresses the scenarios when querying the objective function is expensive and time-consuming, which potentially has many real-world applications.
2.	The paper is clearly structured with detailed descriptions. The authors clearly explain the motivations, challenges, the details of the datasets, etc.
3.	The codebase looks neat. The README looks easy to follow, and the code seems to have sufficient comments with the design of base classes.

**Weaknesses:**

1.	The design of the oracle function is confusing to me. The authors note that “use models built by domain experts as our ground-truth oracle functions… the training data provided for offline MBO is still real data”. Sim2Real itself is a very challenging problem. It is unclear to me why we should optimize based on the real data and evaluate with the models built by domain experts. The evaluation protocol seems questionable due to the gap between simulation and real data. Putting it another way, if an algorithm can perform well on the “real oracle” by using real data, how can we make sure it also performs well on the designed oracle function.
2.	The toy dataset in Figure 1 seems not realistic. The authors note that “but discard the samples that have the combination of best x and y.” In real-world applications, it seems not reasonable to mask some good samples. Rather, we may sample data following a distribution, such as uniform distribution. I am curious whether the offline black-box model-based optimization can discover better design in this setting.

**Additional Feedback:**

A typo:

Line 47, To address -> To address this


**Documentation:**

The documentation is sufficient, with detailed instructions in README and comments in the API.

**Ethics:**

NaN

**Relation To Prior Work:**

The authors discussed sufficient related work

**Summary And Contributions:**

The paper presents a benchmark for offline black-box model-based optimization. The goal is to find an input to maximize an unknown objective function based on a collected dataset. The motivation is that querying the objective function could be expensive in practice, and one prefers to use the data one already has. This paper collects some previous datasets and presents a suite of tasks as a benchmark, including problems in biology, materials science, and robotics. The authors further benchmark many existing algorithms. The code is also open-sourced on GitHub under the MIT license.

---

> ### Author Response · Authors · 2021-07-15
> **Response to Reviewer nw61**
>
> We thank the reviewer for the detailed comments and constructive suggestions. We address the reviewer’s concerns below.
>
> ____
>
> **Non-exact oracles.** We agree with the reviewer that building a good oracle model is challenging, and certainly the discrepancy between the ground truth and our oracle models is a valid concern. We do want to emphasize however, that the goal of our benchmark is not to solve these specific tasks in the real world. For example, we include the GFP task not to help biologists designing better fluorescent proteins. Instead, the goal of this benchmark is to systematically evaluate offline MBO algorithms in a diverse set of tasks. Therefore, while it is possible that an algorithm that performs well on one task might not work as well under the real-world evaluation, it is very unlikely that an algorithm that performs well under all tasks -- which include tasks that do not have this gap at all and exact computation of the objective function is possible -- would be a bad algorithm in the real world. One the other hand, we do try to close the gap between the oracle and the real-world. Our expert model oracles are all trained on more data than the offline MBO algorithms can use, and the majority of our expert model oracles achieve a high rank correlation with the ground truth in a held out validation set. We already include more detailed analysis of the fidelity of our learned oracle models in Appendix F, which shows that different oracles tend to agree on the relative performance between algorithms, which further justifies the use of expert model oracles.
>
> ____
>
> **Illustrative toy MBO task in Figure 1.** Note that the intended use of the illustrative MBO task in Figure 1 is to provide a didactic scenario in which offline MBO methods can actually improve over the training set. Specifically, the only reason why we are masking out good examples in this illustrative task is to mimic real scenarios where better solutions exist in the vicinity of the training data available -- this is the case with many practical problems (e.g., for the task HopperController-Exact-v0, where the optimal controller is not contained in the full unobserved task dataset).
>
> We also note that while in this 2D example, uniform sampling is likely to solve the task (since it is extremely easy to cover the space via uniform sampling), we can easily scale up the example to higher dimensions, at which point, it is exponentially less likely for uniform sampling to provide us with near optimal solutions. Such a scenario will still benefit from the principle of compositionally, and offline MBO will still find a better solution.  We reran the toy task with 20 dimensions **without masking out any top performing examples**, and [we present the result of the same naive gradient ascent offline MBO algorithm here](https://imgur.com/a/HT8lEJr). We see that in even 20 dimensions, the combination of all good options are rare, and the offline MBO algorithm can leverage the compositionally to find a solution way better than any examples in the dataset.

---

### Official Review · Reviewer_VHeZ · 2021-07-05
**Well motivated work but riddled with many flaws.**

**Rating:** 3
**Confidence:** 4

**Strengths:**

The motivation of the benchmarks is well justified.

The initial example of how offline MBO can find designs better than the best in the dataset is appreciated.

Code on GitHub is made available and looks relatively clean.


**Weaknesses:**

The choice of tasks are questionable in several regards:

1. The Hopper task, together with Ant and D’Kitty Morphology tasks, are classic tasks in reinforcement learning (RL) whose objective functions are computationally cheap to query. This makes them bad candidates for offline MBO settings. In fact, the RL community routinely performs online MBO in this type of task.

2. Non-exact oracles can be problematic. Of the 5 remaining tasks (excluding the 3 mentioned in the previous point), 4 of them use non-exact oracles. Using a trained model as an oracle is generally problematic, since these oracle models will make wrong predictions and will suffer from common issues such as distribution shift. In addition, the ResNet oracle for ChEMBL (rho=0.3208) is simply bad to begin with.

3. The choice of K=128 is not well justified. The authors cite reference 7,10,22 to justify the use of 100 percentile values but not the specific choice of K. I believe that K is an important hyperparameter for the benchmark for two reasons. (1) changing K will likely change the results and/or relative ranking of benchmarked methods. (2) in real world settings, the query time scales linearly with K. If a single query is expensive, one might prefer a model that performs well with small Ks. In summary, Different Ks should be explored and K=128 should be justified.

4. As an additional piece of evidence for poor task/experimental design, results for a few tasks are non-discriminative. For example, 7 out of 9 models score 0.864-0.865 in the GFP task, and results of all 9 models cluster around 0.650 in the UTR task.

MINs results are missing. Although the authors added a note to explain, this suggests the work is not well planned, especially considering that MINs is a model proposed by (a subset of) the authors previously.

Misc.: The paper has some typos such as ones in line 365: “unofmralized” should be “unormalized”, and “50%ile” is unconventional.


**Additional Feedback:**

While I believe that the field of offline MBO would benefit from a high quality benchmark suite, I think many aspects of this work are fundamentally flawed. I would like to suggest that the authors carefully rethink the choice of tasks, oracles, as well as metrics.

**Clarity:**

Overall the writing is clear with one major exception.

It is unclear what metric is being reported in the main table (Table 2). The first paragraph of section 7 describes the numbers as “the 100th percentile performance among them averaged over 8 independent runs”. One need to go to appendix C1 to see that the “performance” is a normalized relative performance bounded between 0 and 1 with higher values indicating better performance. In this regard, the main body is not self-contained.

**Correctness:**

As discussed before, the correctness of non-exact oracles is questionable, the choice of K is not well justified, and thus results can’t be trusted.

**Documentation:**

The documentation on GitHub is fair, basic installation instructions and API examples are included. As of July 4th, no instructions/example commands to reproduce the main results in the paper have been made available at https://github.com/brandontrabucco/design-bench.

**Relation To Prior Work:**

The authors clearly discussed how this work relates to previous contributions. It is a novel type of benchmark.

**Summary And Contributions:**

The paper proposes a set of offline model-based optimization(MBO) datasets and evaluation methods for offline MBO model benchmarking. The tasks are derived from real-world problems in biology, material science, and robotics. The paper also benchmarked 9 popular offline MBO methods and reported results.

---

> ### Author Response · Authors · 2021-07-15
> **Response to Reviewer VHeZ**
>
> We thank the reviewer for the detailed comments and constructive suggestions. We have added additional experiments  including an ablation of the sensitivity of the benchmark to the evaluation budget ‘K’. And we have filled in the missing result for MINs and updated the paper now.
> ____
>
> **Ant, DKitty and Hopper tasks:** For the Ant and D’Kitty Morphology tasks, these tasks are different from traditional reinforcement learning tasks, as they aim at designing robot morphologies instead of a controller. Morphology design has however been studied in the domain of online MBO [1] making them suitable candidates for this benchmark. Moreover, we note that the oracles can **not** be cheaply computed on these tasks: evaluating the performance of an algorithm requires training an RL agent from scratch for an agent with the morphology predicted by the method, and hence is quite expensive (12 hours on a GPU).
>
> We agree with the reviewer that the HopperController task is a classical task in reinforcement learning and can be solved with typical RL algorithms in the online setting. However, this task in this benchmark is different -- rather than deciding the action at each time-step in a trajectory like in RL, we directly aim to learn the weights of a neural network controller. This provides a high-dimensional, challenging task for offline MBO, and as shown in Table 2, most algorithms struggle on this task.
>
> ____
>
> **Non-exact oracles:** We agree with the reviewer that using non-exact expert models as oracles is less than ideal. We do want to emphasize however, that the expert model oracles we use are trained on **(1)** more data than the offline MBO algorithms can use, and **(2)** the majority of our expert model oracles achieve a high rank correlation with the ground truth in a held out validation set. We already include more detailed analysis of the fidelity of our learned oracle models in Appendix F. For the ChEMBL task, we will modify the task to utilize an oracle with much improved performance.in the next revision of the benchmark.
>
>  While we agree with the reviewer that using oracle models that are themselves learned (rather than ground truth) presents a number of challenges, we believe there are few viable alternatives for benchmarking MBO methods. We can't evaluate methods comparatively using real-world experiments, as this would be prohibitively expensive, so there are only two choices: **(1)** domain-specific simulation models; **(2)** data-driven learned models that use significantly more data. Both have their downsides, and we include both of them in our evaluation: the morphology and controller optimization tasks use domain-specific physics simulation, and the physical and biological tasks use learned oracles trained on much larger datasets, as these domains do not have readily available and accessible domain-specific simulators that we could use.
>
> ____
>
> **Value of K.** We initially chose K=128 as a default choice, which was not tuned in any manner.. We also conducted additional ablation studies regarding different K values and we present them below. For all the tasks in the benchmark, we evaluate all the methods with K ranging from 2 to 512, and we compute the aggregated rank correlation of the ranks of different offline MBO algorithms benchmarked in the paper between different Ks. [We show the results in this heatmap](https://imgur.com/a/aJvPu6f). We see that overall the correlations are high between all Ks, especially for K > 16, showing that the benchmark is not sensitive to the choice of K. We will also include this result in the final version of the paper.
>
> ____
>
> **Non-discriminative tasks.** While we agree with the reviewers that some tasks in this benchmark are less discriminative, we would also like to point out that most other tasks (all except UTR and GFP) are discriminative. We included UTR and GFP largely because prior offline MBO works [2, 3] have used these tasks for benchmarking and this allows for backward compatibility. We intend that these tasks can serve as a sanity check for any new algorithm designer.
>
> ____
>
> **Documentation.** Regarding reproducing the experiments in the paper, the implementation of all the offline MBO algorithms is released in a separate GitHub repository, https://github.com/brandontrabucco/design-baselines, which is also mentioned in Section 6 of the paper, and also already [directly linked in our main design-bench GitHub repository](https://github.com/brandontrabucco/design-bench#reproducing-baseline-performance).
>
> **References**
>
> [1] Liao, Thomas, et al. "Data-efficient learning of morphology and controller for a microrobot." 2019 ICRA. IEEE, 2019.
>
> [2] Angermueller, Christof, et al. "Population-based black-box optimization for biological sequence design." ICML. 2020.
>
> [3] Brookes, David, Hahnbeom Park, and Jennifer Listgarten. "Conditioning by adaptive sampling for robust design." ICML 2019.

---

### Decision · Program_Chairs · 2021-07-26

**Decision:**

Reject

**Comment:**

The paper proposes a benchmark for model-based optimization (MBO). Reviewers praised the submission as well-motivated and well-written, and agreed that such a benchmark could be useful; reviewers also found the codebase to be clean and well-documented. However all reviewers raised concerns about the choice and design of the particular tasks included in the benchmark, and most notably about the use of non-exact oracle functions.

Of the eight tasks included in the benchmark, only four (Ant Morphology, D’Kitty Morphology, Hopper Controller, and TF Bind 8) have exact oracle functions. Reviewer VHeZ claims that the first three are standard tasks in RL; this is not exactly correct, since the standard RL formulations of these problems involve learning a policy to control a fixed morphology. However it is not clear that the formulations of these tasks are realistic. The Ant and D’Kitty Morphology tasks are to find a robot morphology that maximizes the performance of a fixed and pretrained controller; however it is hard to imagine realistic situations where robot morphology can be changed but the controller is fixed, since the former would require changing hardware but the latter can be achieved in software. The Hopper Controller task seems similarly contrived; while it is distinct from the standard RL task (L241-L244), it is not clear why any real-world learning scenario would give learners access to network weights paired with overall returns and without any trajectory information.

The other four tasks rely on learned oracle functions. All reviewers expressed concern with the use of learned oracle functions, since learned oracles may not precisely match the physical reality of the task. The AC is sympathetic to the author’s reply that the goal of the benchmark is not to improve performance on the underlying tasks but instead as a generic benchmark for MBO algorithms; however even with this motivation the authors should demonstrate that the tasks in the benchmark are reasonable tasks where MBO might be deployed, and that the tasks can indeed be used to differentiate MBO algorithms. From this perspective, the ChEMBL task seems ill-suited since its learned oracle has a very low rank correlation with a held-out validation set, and as Reviewer VHeZ points out the GFP and UTR tasks seem ill-suited since they do not clearly discriminate between the MBO algorithms analyzed.

Overall despite the submission’s clear motivations, concerns with the design of many of the tasks comprising the benchmark make it unclear whether it will truly be useful for the future development of MBO algorithms. The authors are encouraged to take these concerns into account and submit a revised version of the benchmark to a future venue.